# Interplay between DNA damage repair and apoptosis shapes cancer evolution through aneuploidy and microsatellite instability

Noam Auslander[1✉], Yuri I. Wolf [1] & Eugene V. Koonin [1✉]

Driver mutations and chromosomal aneuploidy are major determinants of tumorigenesis that exhibit complex relationships. Here, we identify associations between driver mutations and chromosomal aberrations that define two tumor clusters, with distinct regimes of tumor evolution underpinned by unique sets of mutations in different components of DNA damage response. Gastrointestinal and endometrial tumors comprise a separate cluster for which chromosomal-arm aneuploidy and driver mutations are mutually exclusive. The landscape of driver mutations in these tumors is dominated by mutations in DNA repair genes that are further linked to microsatellite instability. The rest of the cancer types show a positive association between driver mutations and aneuploidy, and a characteristic set of mutations that involves primarily genes for components of the apoptotic machinery. The distinct sets of mutated genes derived here show substantial prognostic power and suggest specific vulnerabilities of different cancers that might have therapeutic potential.

[1] National Center for Biotechnology Information, National Library of Medicine, National Institutes of Health, Bethesda, MD 20894, USA. ✉email: noam. auslander@nih.gov; koonin@ncbi.nlm.nih.gov

Acquisition of genetic alterations is thought to drive the progression of normal cells through hyperplastic and dysplastic stages to invasive cancer and, ultimately, to metastatic disease. In recent years, analysis of the increasingly abundant cancer genomics, transcriptomics and proteomics data has substantially improved our understanding of tumor development through the activation of oncogenes and inactivation of tumor suppressors[1–3]. In addition to driver mutations in oncogenes and/or tumor suppressors, the majority of solid tumors display widespread whole chromosome or chromosome arm imbalances (here termed aneuploidy), as well as large deletions, inversions, translocations, and other genetic abnormalities[4]. Despite the fact that numerical and structural chromosome abnormalities are the most pronounced, distinguishing characteristics of cancer genomes, the role of arm and chromosome level aneuploidy in tumor development remains poorly understood[5–7]. In particular, the genes and pathways that might be affected by aneuploidy remain largely unknown.

Several studies have investigated the relationships between different genetic alterations in cancer and reported an inverse correlation between the number of recurrent copy number alterations and the number of somatic mutations[8,9]. However, more recent work has demonstrated that the correlation between aneuploidy and non-silent somatic mutation rate is actually positive for the majority of tumors, but for several cancer types, including gastrointestinal and endometrial tumors, this correlation is significantly negative[10–12]. *TP53* is the only gene for which the mutation rate has been shown to positively correlate with the aneuploidy level across tumor types, consistent with previous findings[8,10,13]. For other genes, the pan-cancer associations of mutation rates with aneuploidy have been found to be largely negative and less significant[8].

Here we perform a pan-cancer analysis of the interplay between mutations, specifically in cancer driver genes and chromosomal-arm level aneuploidy, and its consequences for clinical outcome. In similar to the reported for mutation load and all aneuploidies[10–12], we find positive correlations between chromosomal arm level aneuploidy and driver mutations load for the majority of cancers, but in gastrointestinal and endometrial tumors, the correlation is strongly negative. The latter cancers also show an unexpected association of driver mutations with improved overall survival rate. Identification of unique mutational gene sets shows that, in the two clusters of tumors, the load of driver mutations is associated with distinct DNA Damage Response (DDR) pathways. In gastrointestinal and endometrial tumors, high load of oncogenic mutations is predominantly observed in tumors mutated in DNA repair genes, whereas in the other tumor types, high load of oncogenic mutations corresponds to apparent inactivation of the apoptosis network and DNA damage checkpoints. The ratio of the mutation load in the DNA repair system to that in the checkpoint and apoptotic machinery is shown to be a pan-cancer correlate of aneuploidy and overall survival which subdivides tumors into two major classes. In the first class, tumorigenesis appears to be driven, primarily, by mutations in repair genes that allow mutations to accumulate increasingly but preclude chromosomal aberrations. In the second class of cancers, tumor development is apparently driven by mutations in DNA damage checkpoint and apoptosis genes which allow uncontrolled cell division accompanied by diverse chromosomal alterations. For the first class that consists of gastrointestinal and endometrial tumors, we additionally derive a mutational gene set that captures the mutual exclusivity between aneuploidy and microsatellite instability (MSI). This set reflects differences in therapeutic vulnerabilities and can be used as an independent prognostic marker within this tumor class. Overall, our analysis reveals genomic determinants of aneuploidy and clinical outcome, uncovering their relations with driver mutations and distinguishing DDR pathways that appear to promote tumor development through separate courses.

## Results

**Associations of driver mutations, aneuploidy, and survival.** For the purpose of this analysis, we integrated mutational, aneuploidy, and clinical data from 8686 tumor samples from 32 solid tumor types represented in The Cancer Genome Atlas (TCGA)[11,14] (Table 1, Supplementary Data 1). First, we analyzed the correlation between the number of mutations[15] in cancer driver genes (which is used as a proxy for the number of actual driver mutations) and aneuploidy levels in each tumor type. In agreement with the previous observations for the overall mutational load[10,11], the correlations were positive for most tumor types, but significantly negative for gastrointestinal and endometrial tumors in which we also noticed a higher load of driver mutations (Fig. 1a). We next investigated the association between the number of driver mutations and overall survival rates. We found that, although in most tumor types, a large number of driver mutations is predictably associated with poor outcome, most of the gastrointestinal and endometrial tumors show an inverse relationship (Fig. 1a, b). This trend is recapitulated with aggregated data from these two classes for tumor types although the different survival rates in different tumor types are likely to be a confounding factor in this analysis (Fig. 1c). However, the overall mutational burden is positively correlated with survival rates mostly in hypermutated tumors (including those with a negative association between driver mutations and survival, such as lung carcinomas; Supplementary Fig. 1), consistent with previous findings[16]. Similar associations are observed when using Poly-Phen[17] and SIFT[18] scores to predict functional alterations in driver mutations (Supplementary Fig. 2). Furthermore, these associations are reproduced when controlling for the total mutation burden and when considering whole chromosome aneuploidy or separately analyzing arm gains and losses. Together, these observations further support the unique associations characteristic of gastrointestinal and endometrial tumors (Supplementary Figs. 3, 4). We also examined the associations between focal Somatic Copy Number Alteration (SCNA) levels and driver mutations for the 13 tumor types with available focal SCNA data[12] and found that these do not necessarily fully conform with the pattern observed for whole chromosome aneuploidy or arm gains and losses (Supplementary Fig. 4). A more complete analysis of focal SCNA remains to be performed. The unexpected, complex relationship between the load of driver mutations, arm-level aneuploidy and patient survival partitions tumors into two classes: one in which different types of genetic alterations are positively correlated and appear to jointly account for poor survival, and a second one where these events are observed in distinct tumors, such that aneuploidy seems to be uniquely associated with poor prognosis (Fig. 1d, Supplementary Fig. 5).

Driver mutations and aneuploidy are genetic alterations that are facilitated by genome instability, which is are a result of impairment of DNA Damage Response (DDR)[19–21]. Consequently, we next sought to identify the specific DDR pathways that are affected in the two tumor classes, and to investigate how impairment of different forms of DDR might promote tumor evolution through alternative routes.

**Distinct DDR pathways characterize the two tumor classes.** Starting with a set of 746 genes implicated in DNA damage response (DDR) pathways (based on Gene Ontology annotations[22], Supplementary Data 2), we aimed to derive distinct

**Table 1 TCGA pan-cancer datasets.**

| Tumor type | TCGA ID | Number of cases | Cases with MSI |
|---|---|---|---|
| Pancreatic adenocarcinoma | PAAD | 161 | |
| Sarcoma | SARC | 227 | |
| Cholangiocarcinoma | CHOL | 36 | |
| Lymphoid neoplasm diffuse large B-cell lymphoma | DLBC | 37 | |
| Prostate adenocarcinoma | PRAD | 469 | |
| Lung squamous cell carcinoma | LUAD | 495 | |
| Liver hepatocellular carcinoma | LIHC | 349 | |
| Lung adenocarcinoma | LUSC | 463 | |
| Testicular germ cell tumors | TGCT | 128 | |
| Bladder urothelial carcinoma | BLCA | 401 | |
| Adrenocortical carcinoma | ACC | 89 | |
| Kidney chromophobe | KICH | 65 | |
| Kidney renal clear cell carcinoma | KIRC | 344 | |
| Breast invasive carcinoma | BRCA | 757 | |
| Glioblastoma multiforme | GBM | 292 | |
| Skin cutaneous melanoma | SKCM | 456 | |
| Pheochromocytoma and paraganglioma | PCPG | 160 | |
| Thyroid carcinoma | THCA | 458 | |
| Ovarian serous cystadenocarcinoma | OV | 61 | |
| Thymoma | THYM | 105 | |
| Mesothelioma | MESO | 79 | |
| Cervical squamous cell carcinoma and endocervical adenocarcinoma | CESC | 278 | |
| Kidney renal papillary cell carcinoma | KIRP | 273 | |
| Head and neck squamous cell carcinoma | HNSC | 491 | |
| Brain lower grade glioma | LGG | 502 | |
| Esophageal carcinoma | ESCA | 162 | |
| Uveal melanoma | UVM | 84 | |
| Rectum adenocarcinoma | READ | 80 | 3 |
| Uterine carcinosarcoma | UCS | 56 | 0 |
| Stomach adenocarcinoma | STAD | 423 | 49 |
| Colon adenocarcinoma | COAD | 278 | 53 |
| Uterine corpus endometrial carcinoma | UCEC | 427 | 118 |

brevity, apoptosis set, Supplementary Fig. 6), primarily, *TP53* and the associated apoptosis and checkpoint factors, such as *BCL3, BRCA2, CHEK2, PML, TOPORS, TP63, AEN*, and *SIRT1*, which are involved in the P53-dependent damage response. The mutation loads of these sets show highly significant, positive correlations with the loads of driver mutations across tumors in the respective class (Fig. 2b, which is not observed for most of the randomly chosen sets of mutation; Supplementary Fig. 7). The unique identities of the two mutated gene sets are further corroborated by the observed distinctive, highly significant enrichment of multiple DDR pathways (according to GO) with genes from the respective sets (Fig. 2c, Supplementary Fig. 6). Crucially, these sets show a pan-cancer correlation with the clinical outcome, whereby a high ratio of apoptosis to repair mutations strongly correlates with lower overall survival (Fig. 2d), as well as lower progression-free interval and disease-specific survival (Supplementary Fig. 8).

The two mutation sets show opposite associations with aneuploidy across cancers: 30 of the 32 tumor types exhibit positive correlations between aneuploidy and the apoptosis set mutations count, of which 13 were significant (Spearman rank-correlation *P*-value < 0.05), whereas the correlations between aneuploidy and the repair set mutation count were negative for 25 cancer types, and significant for 5 of these (Fig. 3a). The negative association of aneuploidy with the repair mutation set is most pronounced in gastrointestinal and endometrial tumors (right end of the spectrum in Fig. 3a), whereas the positive association with the apoptosis set is mainly manifested at the left end of the spectrum that includes tumors with lower loads of driver mutations (Fig. 1a). The ratio of the mutation load in the apoptosis set to that in the repair set positively correlates with aneuploidy for nearly all tumor types (and significantly for 18 of these, Fig. 3a), with the exception of brain lower grade glioma (LGG). Thus, samples with a higher load in the repair set, as opposed to the apoptosis set, show significantly elevated aneuploidy levels across cancers (Fig. 3b). Moreover, the samples with the highest ratio (top 5%) of apoptosis to repair set mutations completely lack mutations in the repair set genes (whereas the samples with the lowest ratio carry mutations in both sets, Fig. 3c). Indeed, *TP53* shows the strongest positive association with aneuploidy as the only gene that is positively and significantly associated with aneuploidy in gastrointestinal and endometrial tumors (Supplementary Fig. 9). Nevertheless, excluding *TP53* (as well as *BRCA2*) from the apoptosis mutated gene set does not eliminate the association of the ratio between the repair and apoptosis sets with aneuploidy (Supplementary Fig. 10)

mutational gene sets of DDR genes that might be associated with the load of driver mutations in each of the two tumor classes. To this end, we applied 100 repetitions of a genetic algorithm for every tumor type and calculated a selection score for each DDR gene, along with the corresponding binomial *P*-value (see Methods for details). Genes with a significant combined P-value for one of the two classes of cancer types (Fisher *P*-value < 0.1 for a single class) were selected, and two distinct mutational gene sets of DDR genes were derived, each uniquely associated with one of the two tumor classes (Fig. 2a). The mutational gene set for the gastrointestinal and endometrial tumors predominantly includes DNA repair genes (hereafter repair set), in particular, base excision repair (*XRCC1* and *XRCC6*), nucleotide excision repair (NER, *ERCC1-6*), mismatch repair (*MSH2-4* and *MSH6, MLH1, and MLH2*), non-homologous end joining (*PARP1* and *BRCA1*), and homologous recombination (*RAD51, XRCC2*, and *XRCC3*). In contrast, the mutated gene set for the second, larger tumor class encompasses numerous genes involved in DNA damage checkpoints and damage-induced apoptosis (hereafter, for

**Associations between MSI, aneuploidy, and clinical outcome.** Our analyses show that gastrointestinal and endometrial cancers form a separate class of tumors in which aneuploidy is anticorrelated with the load of driver mutations. Furthermore, these tumors are characterized by predominant mutations in DNA repair genes and a paradoxical, inverse dependency between driver mutations and survival. Additionally, a subset of tumors in this class shows high MSI. Similarly to the previous findings for colorectal tumors[23,24], we demonstrate an inverse association between MSI and aneuploidy across all gastrointestinal and endometrial TCGA tumors, and a positive association between MSI and driver mutations except for those in *TP53* and *APC* genes (Supplementary Fig. 9). Accordingly, we derived a third mutated gene set to represent the apparent mutual exclusion between aneuploidy and MSI in these tumors, so that to simultaneously maximize the positive association with MSI and the negative association with aneuploidy, focusing on DDR and

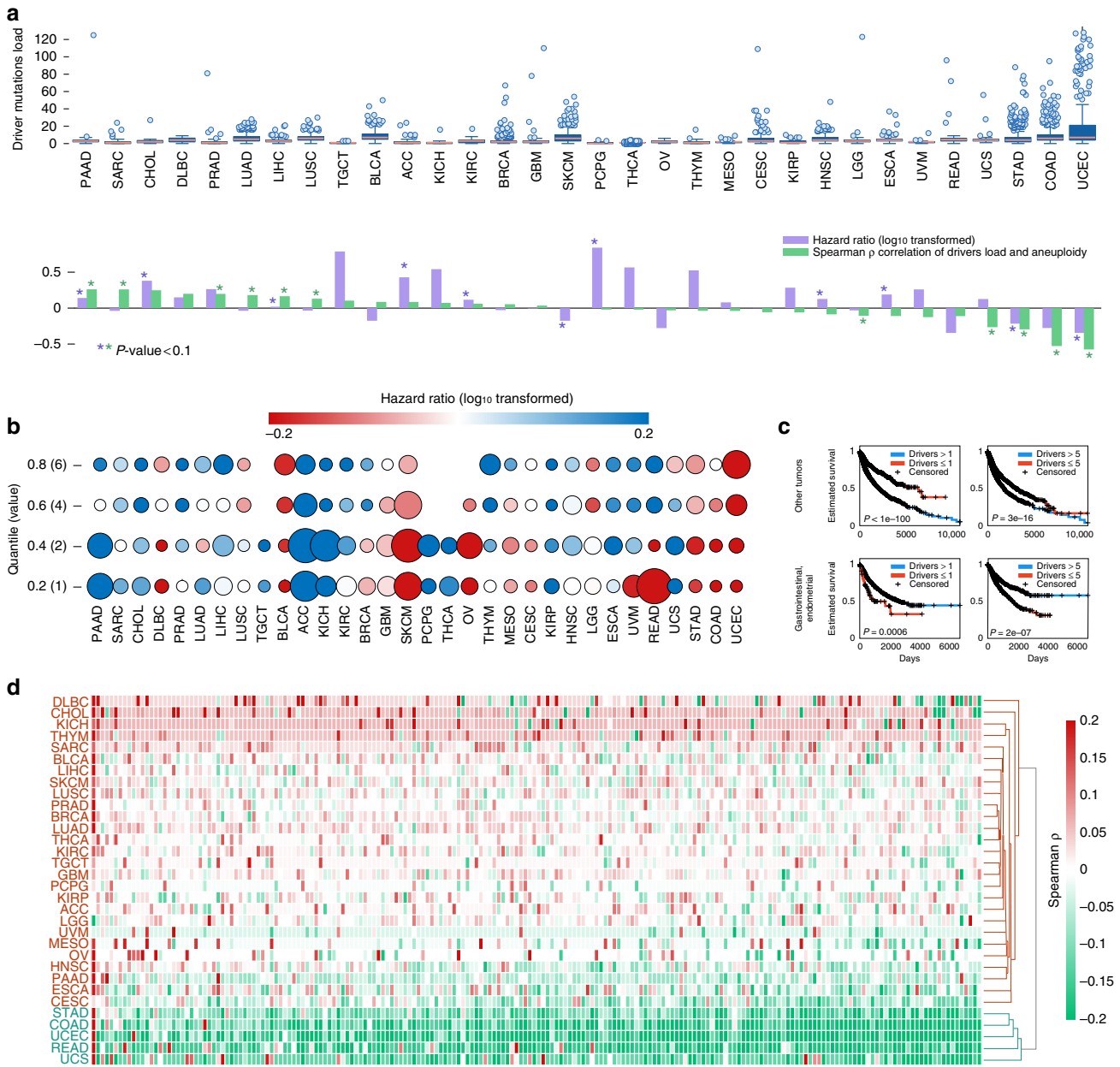

**Fig. 1 Pan-cancer association between the number of driver mutations, levels of aneuploidy and overall survival. a** Top panel: Boxplots showing the distribution of the number of driver mutations per sample in each tumor type. Center lines indicate medians, box edges represent the interquartile range, whiskers extend to the most extreme data points not considered outliers, and the outliers are plotted individually. Bottom panel: the corresponding correlation coefficients between the number of driver mutations and aneuploidy scores, and the hazard ratio values (log10 transformed) resulting from Kaplan–Meier overall survival curves for samples with high vs. low number of driver mutations (separated by the median). Positive log10-hazard ratio values indicate that high load of driver mutations is associated with worse survival, and negative log10-hazard ratio values indicate that high load of drivers is associated with improved survival. Statistical significance (log-rank and Spearman rank-correlation *P*-value < 0.05) is indicated with asterisk. **b** The hazard ratio values resulting from Kaplan–Meier overall survival prediction curves for samples with high vs. low number of driver mutations for different thresholds (y-axis), for different tumor types (x-axis). The circle sizes represent the significance level measured as log-rank *P*-value. **c** Kaplan–Meier curves predicting overall survival for gastrointestinal and endometrial tumors (bottom panels) and for all other tumors (top panels), for tumors with high vs. low number of cancer driver mutations separated with two thresholds. The log-rank *P*-values are indicated. **d** tumor clustering based on the associations between aneuploidy and driver mutations (columns) for each tumor type (rows). Source data are provided as a Source Data file.

cancer driver genes (using a feature selection process similar to that employed for the other sets; see Methods for details). The 17 genes in the selected optimal set reflect the tradeoff between aneuploidy and MSI in gastrointestinal and endometrial cancers (Fig. 4a, b), and are strongly enriched in mismatch repair and double strand break repair genes (*MLH1, MSH2, PMS2, DNA2,*

*FBXO18, RAD21,* and *RPA1*). The MSI-aneuploidy set was highly predictive of MSI not only in the TCGA data on which it was trained, but also in two independent test data sets for colorectal adenocarcinoma (COADREAD, Receiver operating characteristic Area Under the Curve (AUC) = 0.85 and 0.95), one test data set for stomach adenocarcinoma (STAD, AUC = 0.85), and one for

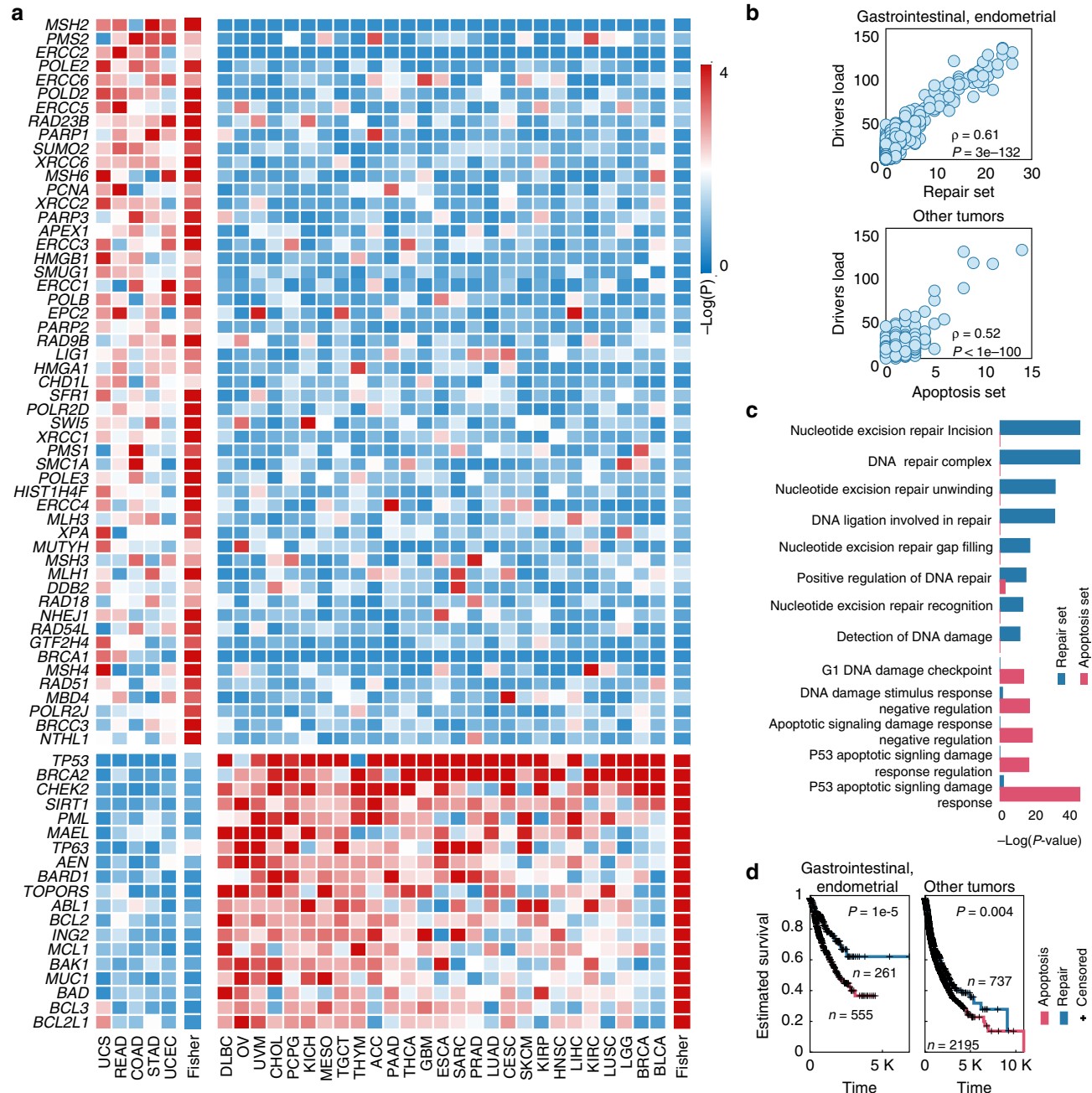

**Fig. 2 Driver-associated mutation sets distinguish DNA repair from damage-induced apoptosis. a** Heatmaps showing selection P-values (negative log-scaled) assigned to each gene in the selected mutational sets (rows) for every tumor type and the combined P-values for each cluster of tumors (columns). **b** scatter plots correlating the driver mutation load (y-axes) with the repair set load (upper panel x-axis, for gastrointestinal and endometrial tumors), and with the apoptosis set load (bottom panel x-axis, for all other tumor types). **c** hyper-geometric enrichment P-value for the DDR pathways enriched with genes from one of the selected sets. **d** Kaplan–Meier curves predicting overall survival for patients with higher repair set mutation rate (i.e. more repair set mutations than apoptosis set mutations, blue) vs. those with higher apoptosis set mutation rate (red), for gastrointestinal and endometrial tumor samples (left panel) and all other tumor samples (right panel).

uterine corpus endometrial carcinoma (AUC = 0.92, Fig. 4c). The high mutation load of this set is associated with better survival across the integrated patient cohort of all gastrointestinal and endometrial tumors (Fig. 4d), and individually in each tumor type excluding rectal adenocarcinoma (READ), where the sample size is likely to be a confining factor (Fig. 4e).

MSI has been associated with improved survival in gastrointestinal and endometrial tumors[25–27], whereas chromosomal instability has been linked to poor survival[28,29]. Because of the

strong inverse associations observed between aneuploidy and MSI, we next explored the individual contributions of MSI and aneuploidy to the overall survival, compared with microsatellite-stable (MSS) diploid tumors. We found that aneuploid tumors are associated with the worst outcome, whereas no significant differences were observed between MSS and MSI diploid tumors (Fig. 5a). These findings imply that the favorable outcome associated with MSI could be only due to the diploid karyotype nature of the MSI tumors, as opposed to being caused by MSI as

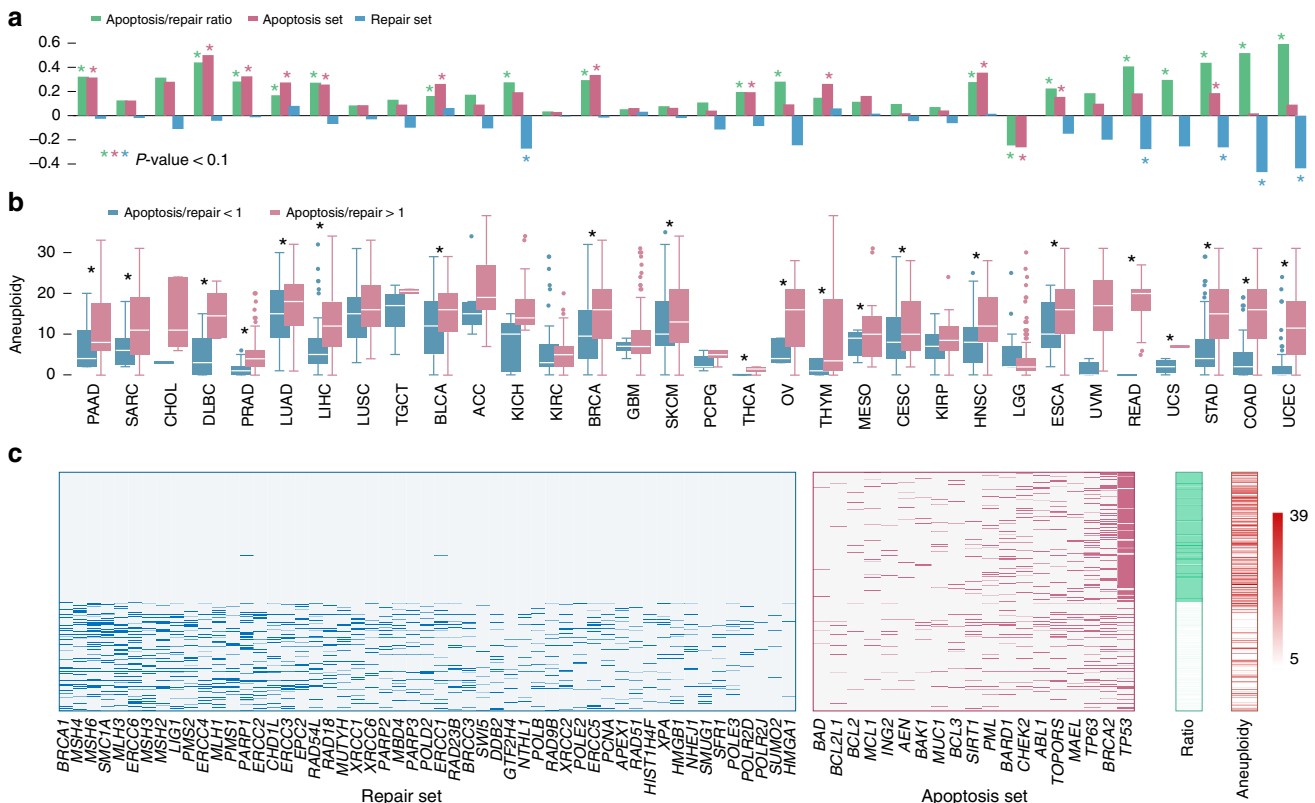

**Fig. 3 DNA repair and apoptosis mutational sets correlate with aneuploidy. a** Spearman correlation coefficients between aneuploidy and the apoptosis and repair set load, and their ratio. Statistical significance (Spearman rank-correlation P-value < 0.05) is indicated with asterisk. **b** Boxplot comparing the aneuploidy distribution between low (<1) and high (>1) apoptosis to repair set load ratio. Statistical significance (Rank-sum P-value < 0.05) is indicated with asterisk. Center lines indicate medians, box edges represent the interquartile range, whiskers extend to the most extreme data points not considered outliers, and the outliers are plotted individually. **c** Map of the repair and apoptosis set mutations and levels of aneuploidy, for samples with highest and lowest apoptosis to repair set load ratios (top and bottom 0.05 quartiles).

such. Supporting this notion, we found that MSI mutational signatures, which expectedly exert negative associations with aneuploidy, are associated with poor survival in diploid tumors (COSMIC signatures 6, 14, 15, 20, 21, and 26[30,31], that have been associated with MSI[31,32]; see Supplementary Fig. 11). The MSI-aneuploidy set mutation load was associated with improved survival independently of MSI and aneuploidy, supporting the considerable prognostic power of the mutations in this set (Fig. 5b, c). To find out whether these survival correlates of MSI and aneuploidy reflect differences in therapeutic vulnerabilities, we obtained chemotherapy response data for this cluster of TCGA tumor types. MSI has been previously linked to improved prognosis, but also has been proposed as a marker of non-response to chemotherapy[26,33–35]. Indeed, we found the rate of complete or partial response to chemotherapy to be considerably higher among MSS compared to MSI tumors, and for diploid vs. aneuploid tumors (Fig. 5d). In accord with these observations, mutation load of the MSI-aneuploidy set was higher in responders for some chemotherapeutic agents (Fig. 5e). Single mutations in the MSI-aneuploidy set were not highly predictive of the response to multiple chemotherapeutic agents, so that the set load performed better than individual mutations (Fig. 5f).

## Discussion

Although aneuploidy is a pervasive characteristic of cancer cells, the molecular basis of aneuploidy and implications for patient prognosis are not well understood for most cancers[36,37]. Here, we partition tumor types into two classes showing opposite

associations of driver mutations with aneuploidy and patients survival. These association patterns reflect distinct sets of mutations in different DDR pathways (Fig. 6). Specifically, the mutated gene set for the gastrointestinal and endometrial tumors that are characterized by a negative association between the driver mutation load and aneuploidy consists primarily of various DNA repair genes (this association is not limited to MSI tumors; Supplementary Fig. 12). In this class of tumors, we also observed a paradoxical, negative association between the driver mutation load and patient survival. Conceivably, this could be caused by the multiple mutations in repair genes that introduce a vulnerability to DNA damage. In contrast, for the rest of the analyzed tumor types, where the association between driver mutations and aneuploidy is positive, the mutational set is dominated by genes encoding components of the apoptosis machinery and DNA damage-related cell cycle checkpoints. An additional facet of cancer genome instability is the inverse relationship between aneuploidy and MSI in the gastrointestinal and endometrial tumors (Fig. 6), for which we derived a third mutational set that was highly enriched in a distinct set of repair genes. Crucially, the mutational sets derived here strongly correlate with patient survival. In particular, the ratio of the mutation loads for the apoptotic set to that of the repair set was found to be a universal correlate of survival: a high ratio corresponds to poor survival, suggesting that this ratio could have a prognostic value.

Our pan-cancer analysis connects different genomic aberrations with distinct DDR pathways and segregates gastrointestinal and endometrial tumors into a separate class, where tumorigenesis might be predominantly driven by defects in specific DNA

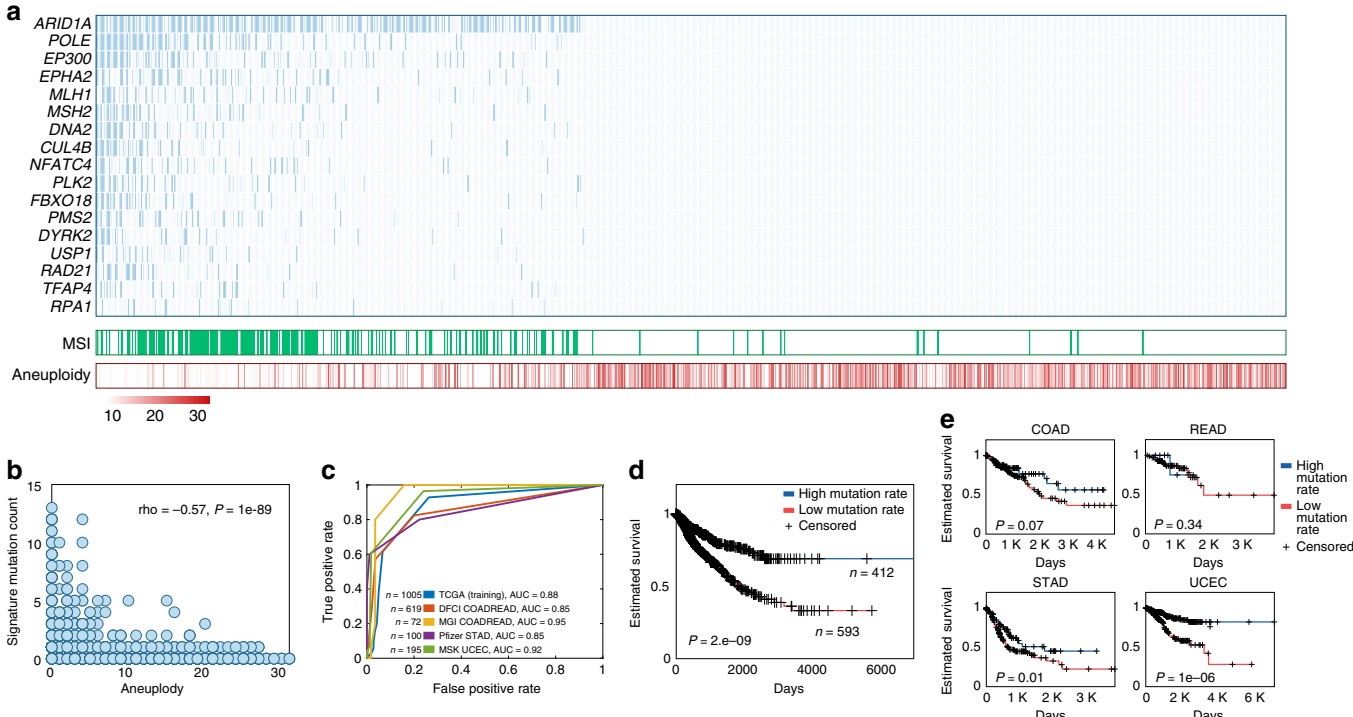

**Fig. 4 Mutated gene set reflecting the tradeoff between MSI and aneuploidy. a** The mutations in the selected set (top panel) positively correlate with MSI (middle panel) and negatively correlate with aneuploidy (bottom panel) across gastrointestinal and endometrial tumor samples (**b**) correlation between aneuploidy (x-axis) and the aneuploidy-MSI set mutation load (y-axis). **c** ROC classification curves predicting MSI using the aneuploidy-MSI set mutation load in the training data and four test datasets. **d** Kaplan–Meier curves predicting overall survival across gastrointestinal and endometrial tumor samples with high vs. low number of aneuploidy-MSI set mutations, separated by the median. **e** Kaplan-Meier curves predicting overall survival in each colon (COAD), rectal (READ), stomach (STAD), uterine corpus (UCEC), and uterine (UCS) carcinomas, for samples with high vs. low number of aneuploidy-MSI set mutations separated by the median. The log-rank P-value are provided.

repair pathways. It has been established previously that mutated apoptosis and DNA damage checkpoint signaling pathways sustain growth with genomic abnormalities[38–40]. The results described here suggest that major deficiencies in DNA repair permit accumulation of oncogenic mutations but not aneuploidy. This finding is in agreement with the inverse relation between MSI and aneuploidy in colorectal tumors[41], and with the diploid karyotype of NER-deficient skin cancer[42]. The inverse relationship between repair and aneuploidy might reflect a direct functional link, whereby intact DNA repair pathways promote the emergence of aneuploidy, or conversely, survival and reproduction of aneuploid cells requires active repair. One possible mechanism underlying such a functional link could be chromatin remodeling. While chromatin relaxation is crucial for DNA repair[43,44], condensed chromatin structure in necessary for chromosomal segregation[45]. Tumor cells that are actively engaged in DNA repair might maintain relaxed chromatin structures that hinder chromosomal segregation and increase aneuploidy, whereas tumor cells with dysfunctional repair systems would preserve condensed chromatin that sustain proper chromosomal segregation and preclude aneuploidy. Indeed, several of the genes in the repair set are involved in chromatin remodeling, such as *ALC1 (CHD1L)*, *PARP1*, and *DDB2*. In particular, *ALC1* is a chromatin remodeling enzyme that relaxes chromatin at early stages of DNA repair[46] through *PARP1* and *DDB2* recruitment[47]. Together with the inclusion of several other chromatin remodeling-associated genes in the repair set (*POLE3, EPC2, BRCC3, HMGA1*, and *HMGB1*) and the MSI-aneuploidy set (*EP300, CUL4B, RPA1, ARID1A*, and *NFATC4*), this could

suggest that inhibition of chromatin relaxation resulting from impairing mutations in DNA repair genes prevents the emergence of aneuploidy in tumor cells.

There is, obviously, a complex relationship between aneuploidy and patient prognosis. Although aneuploidy has been associated with poor patient survival and linked with intrinsic drug resistance[28,48], evidence is accumulating that extreme aneuploidy might also be associated with improved patient outcome[29,49,50]. We demonstrate that aneuploidy is compatible with impairment of apoptotic and DNA damage checkpoint signaling pathways[51–53], but is suppressed by inactivation of DNA repair pathways. Furthermore, although MSI has been previously associated with favorable prognosis, the present findings indicate that this connection could result from the lack of aneuploidy in the MSI tumors rather than from any effect of MSI as such. These findings are in agreement with the observed association of aneuploidy with multi-drug resistance[54,55]. Indeed, the loss of DNA repair sensitizes cells to various drugs that induce DNA damage, whereas cells deficient in checkpoint and apoptotic signaling in response to damage lack this type of vulnerability. In addition, MSI tumors show enhanced immune infiltration levels and improved response to immunotherapy[56,57], whereas the reduced mutational load in gastrointestinal and endometrial aneuploid tumors is likely to restrict the benefits of immunotherapy and several targeted therapies. However, the mutual exclusivity between the loss of DNA repair and aneuploidy raises another possibility, namely, that co-occurrence of aneuploidy with defective DNA repair is lethal, suggesting a therapeutic potential for targeting repair processes in aneuploid cells.

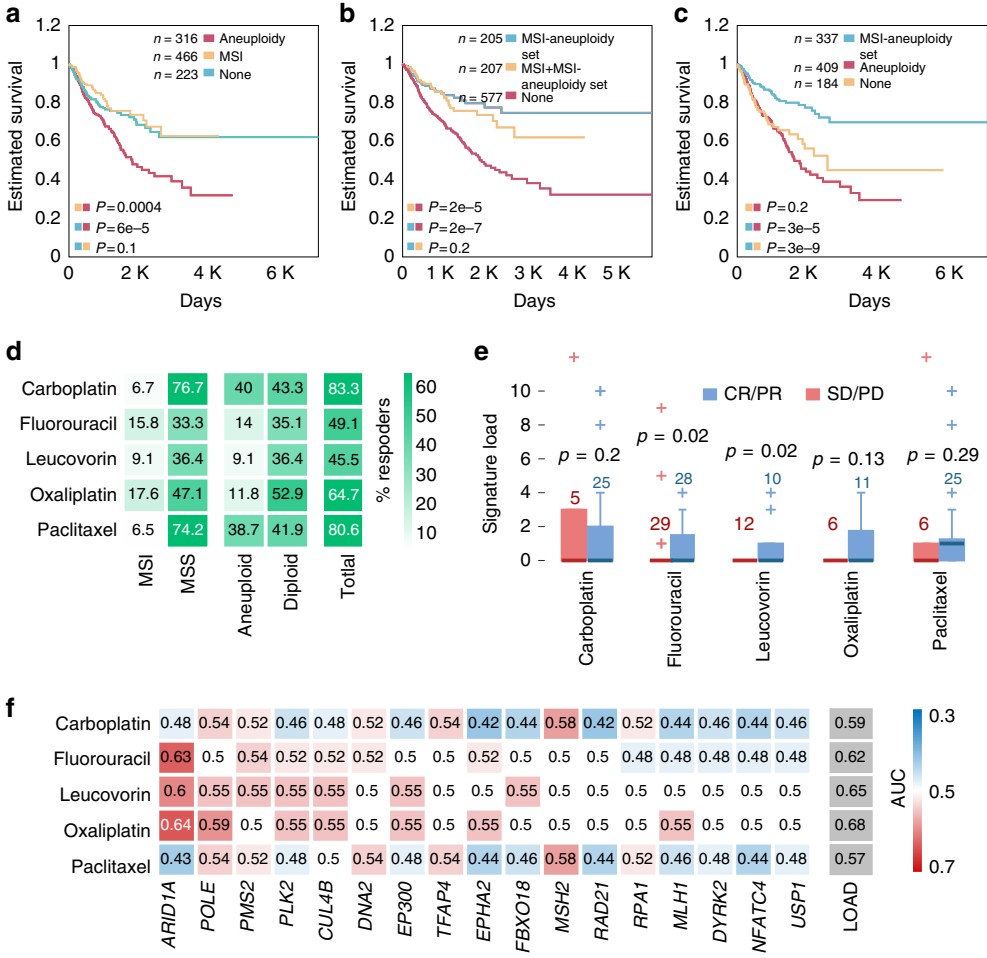

**Fig. 5 Aneuploidy and MSI associations with overall survival and chemotherapy response. a** Kaplan–Meier overall survival curves for MSS tumors with high aneuploidy (red curve, higher than median), MSI tumors (yellow curve), and MSS tumors with low aneuploidy (blue curve, lower than median). **b** Kaplan–Meier overall survival curves for MSS tumors with MSI-aneuploidy set mutations (blue curve), MSI tumors with MSI-aneuploidy set mutations (yellow curve) and MSS tumors with no MSI-aneuploidy set mutations (red curve). **c** Kaplan–Meier overall survival curves for diploid tumors with MSI-aneuploidy set mutations (blue curve), diploid tumors with no MSI-aneuploidy set mutations (yellow curve) and aneuploid tumors with no MSI-aneuploidy set mutations (red curve). The log-rank P-value are provided. **d** Heatmap displaying the percent of responders to each chemotherapeutic agent (rows) for tumors with specific alterations (columns). **e** Boxplots presenting the MSI-aneuploidy set mutation load for responders (blue, CR/PR) vs. non-responders (red, SD/PD), for each chemotherapeutic agent. Center lines indicate medians, box edges represent the interquartile range, whiskers extend to the most extreme data points not considered outliers, and the outliers are plotted individually. **f** AUC map (of ROC classification curves) of individual MSI-aneuploidy set mutations (columns) for predicting response to each chemotherapeutic agent (rows). Source data are provided as a Source Data file.

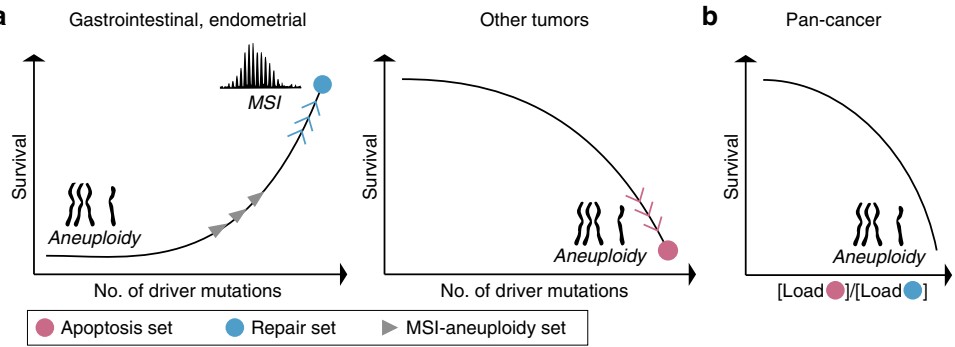

**Fig. 6 A schematic, conceptual depiction of the emerging links between patient survival, driver mutation load and aneuploidy in the two tumor classes. a** Driver mutations, aneuploidy, patient survival, and mutation sets in the two tumor classes. The mutation sets derived to predict high driver mutations load are shown by circles. The inverse relationship between aneuploidy and MSI is demonstrated for the gastrointestinal and endometrial scheme, with the derived MSI-aneuploidy set. **b** The ratio of the mutation loads in the apoptosis vs repair sets as universal predictor of patient survival.

In summary, we reveal here alternative regimes of tumor-igenesis that involve different, either synergistic or mutually exclusive relationships between driver mutations and chromosomal aberrations in different cancer types (Fig. 6). By using large human cancer cohorts, we show that these distinct tumorigenic regimes are underpinned by unique DDR mutational set that appear to govern accumulation of driver mutations and aneuploidy. The derived mutational sets are predictive of patient survival and suggest specific vulnerabilities of different cancer types that might have therapeutic potential.

## Methods

**Data**. TCGA samples of primary and metastatic solid tumors were selected for analysis. The complete mutational data for 32 tumor types in each TCGA study was obtained from the UCEC Xena browser[58], considering all non-silent mutations. Arm-level gain or loss values were obtained for each TCGA sample[11], where the ploidy was determined using the ABSOLUTE algorithm[59]. Each segment was designated as amplified, deleted, or neutral compared with the ploidy of the corresponding sample. The scores assigned to each arm were −1 if lost, +1 if gained, and 0 otherwise. The aneuploidy score for each tumor is calculated as the sum of altered arms, within a range of 0 to 39 (long and short arms for each non-acrocentric chromosome, and only long arms for chromosomes 13, 14, 15, 21, and 22). Sample-wise clinical data was obtained from the TCGA Pan-Cancer Clinical Data Resource (TCGA-CDR[14]). Altogether, 8686 TCGA samples containing all data types including somatic point mutations, aneuploidy scores and clinical data were analyzed (Table 1).

Cancer driver genes were obtained[15], for the drivers analysis a list of pan-cancer driver mutations were used, encompassing the 200 driver mutations[15] that are categorized pan-cancer drivers (i.e. general drivers and not associated with a subgroup of the tumor types, Supplementary Data 3).

Microsatellite Instability (MSI) classification was obtained[60] for uterine corpus endometrial, stomach, colon and rectal carcinomas (UCEC, STAD, COAD, and READ, respectively), for a total of 718 samples of the 8686 with available molecular and clinical data. Drug response data for all drug-patient pairs was obtained[61,62] for these tumor types (considering drugs with sufficient number of samples, $n >$ 15), and categorized into responders (complete or partial response, CR/PR) and non-responders (progressive or stable disease, PD/SD).

To test the MSI-aneuploidy set for MSI prediction in independent test sets, four additional tumor mutational datasets were obtained with MSI classification including two datasets of colorectal adenocarcinoma ($n = 619$[62] and 72[63] with 91 and 15 MSI samples, respectively), one for stomach adenocarcinoma ($n = 100$[64] with 10 MSI samples) and one of uterine corpus endometrial carcinoma ($n = 195$[65] with 28 MSI samples).

**Tumor clustering**. Hierarchical clustering of driver-aneuploidy associations in each tumor type, with the average linkage function and Euclidian distance metric, was performed to classify tumors based on these associations. The clustering was applied to a matrix of correlation coefficients between each driver mutation and aneuploidy (rows) in each tumor type (columns), where missing values (corresponding to missing mutations in the datasets) were assigned the mean correlation value for each tumor type.

**DDR mutational sets predicting driver mutations load**. DDR mutational sets predictive of the driver mutational load were derived for each tumor cluster individually using a Genetic Algorithm (GA) search to produce sets of DDR mutations predictive of drivers load in individual tumor types. For each tumor type, 100 repetitions of the genetic algorithm were run, where the initial population of size of $\frac{n}{4}$ ($n$ is the sample size of the tumor type), was (a) initialized randomly with the $p = 0.05$ probability of each mutation in the population set. The objective set was the Spearman correlation coefficient $\rho$ between the population DDR set load and the driver mutation load, which was (b) evaluated for each item in the population (for its unique set of DDR mutations) on the true population of tumor samples. Then, the top half of the population with the highest $\rho$ with the driver mutation load in the test set was (c) selected for reproduction, where randomly selected pairs from this selected half of the population were chosen for (d) crossover, with $p = 0.05 \times M_i$ probability of mutations in the crossover process ($M_i$ is the number of mutations of item in the population), until a population size of $\frac{n}{4}$ was reached. Twenty iterations of the steps (b–d) were performed, and the best solution (set of mutations with the highest correlation with the load of drivers), was retained. When 100 iterations were completed, the solutions obtained were evaluated to generate a selection score for each DDR mutation $m_i$.

$$\text{Selection score}(m_i) = \sum_{\text{iteration } j} \frac{I_{ij}}{\sum_{\text{mutation } k} I_{kj}}$$

where $I_{ij}$ is the selection of $g_i$ in iteration $j$, thus giving higher weight to $m_i$ that is selected in iterations with fewer selected mutations.

A binomial $P$-value $p_t$ was assigned to each DDR mutation via the resulting scores distribution for each tumor type $t$. For each DDR gene, the $P$-values assigned to tumors in each cluster were combined into test statistic $X^2$ using the Fisher method

$$X^2_{2CT} \sim -2 \sum_{t=1}^{CT} \ln(p_t)$$

where $pt$ is the $P$-value assigned to a mutation in tumor type $t$, and CT is the number of tumor types in a cluster.

The $X^2$ $P$-values were then derived for each mutation in the two tumor classes individually. The final set derived for each cluster consisted of mutations with significant $X^2$ $P$-values (with $\alpha < 0.1$ cutoff) only in the corresponding tumor class (i.e. not significantly associated with the other class; see Supplementary Data 4 and Fig. 13). Using a stricter cutoff did not change the selected genes in the apoptosis set, and yielded 45 of the 53 genes in the repair set, with performance similar to the original one (Supplementary Fig. 14).

Repeating this analysis without limiting the search for enriched mutated gene sets to DDR genes (i.e. starting from all genes) did not yield a DNA-repair enriched set of genes for the gastrointestinal and endometrial tumors. This is likely to be the case because tumors with impaired mismatch repair contain mutations in many different genes, thus making in difficult to identify the initial set of mutated DNA repair genes. By contrast, the set selected for other tumor types was still enriched with genes involved in apoptotic pathways. Crucially, the ratio between these sets showed similar associations with aneuploidy and overall survival rates as the DDR-limited search (Supplementary Fig. 15).

**MSI-aneuploidy set**. To derive a mutational set that would be simultaneously predictive of MSI and low aneuploidy, different sets predicting MSI and low aneuploidy were obtained independently. The genetic algorithm described above was applied to the cluster of gastrointestinal and endometrial tumor samples using (a) 100 repetitions aiming to maximize the Spearman correlation coefficient $\rho$ between each set and the aneuploidy level and (b) 100 repetitions aiming to maximize the performance (AUC of ROC curve) of each set in predicting the MSI status. Mutations significantly selected for both tasks (with combined $X^2$ $P$-value < 0.1 for the selection scores over 100 repetitions for both (a) and (b)) were chosen to compose the final MSI-aneuploidy set.

**Statistical analysis**. Boxplots and comparisons: for all boxplots, center lines indicate medians, box edges represent the interquartile range, whiskers extend to the most extreme data points not considered outliers, and the outliers are plotted individually. Points are defined as outliers if they are greater than $q_3 + w \times (q_3 - q_1)$ or $< q_1 - w \times (q_3 - q_1)$, where $w$ is the maximum whisker length, and $q_1$ and $q_3$ are the 25th and 75th percentiles of the sample data, respectively. All differential expression and distribution comparisons P-values are obtained via one-sided Rank-sum test.

Survival analyses: Kaplan–Meier analyses are performed by comparing the survival of patients with high scores to those with low scores, using a one-sided log-rank test.

Correlation coefficients: correlations coefficients and $P$-values were obtained using the Spearman rank correlation test.

Pathway enrichment analysis: enrichment $P$-values were calculated using the hypergeometric enrichment test, using GO annotation pathway definitions.

**Reporting summary**. Further information on research design is available in the Nature Research Reporting Summary linked to this article.

## Data availability

The TCGA datasets referenced during the study are available from the Xena browser [https://xenabrowser.net] and cBioPortal [https://www.cbioportal.org]. The source data underlying Figs. 1–5 and Supplementary Figs. 1–4 and 7–10 are provided as a Source Data file. All the other data supporting the findings of this study are available within the article and its supplementary information files and from the corresponding author upon reasonable request. A reporting summary for this article is available as a Supplementary Information file.

## Code availability

All code was implemented in MATLAB_R2018a and is publicly and freely available in the GitHub repository:

[https://github.com/noamaus/INTERPLAY-TUMOR-CODES]

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

## Acknowledgements

We thank Koonin group members for helpful discussions. The authors' research is supported by intramural research program funds of the National Institutes of Health (National Library of Medicine). This work utilized the computational resources of the NIH HPC Biowulf cluster. (http://hpc.nih.gov).

## Author contributions

E.V.K. initiated the study; N.A. performed research; N.A., Y.I.W., and E.V.K. analyzed the data; N.A. and E.V.K. wrote the manuscript that was edited and approved by all authors.

## Competing interests

The authors declare no competing interests.

## Additional information

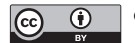

