## [Peer Review File · Nature Communications]

Reviewers' comments:

Reviewer 1:

In this manuscript, Auslander and colleagues analyzed the relationship between aneuploidy and point mutations across tumors. They find that in colorectal, uterine, stomach and few other tumors, aneuploidy is strongly anti-correlated with mutation load. On the contrary, for the other tumor types, aneuploidy and point mutations are positively associated. They go on to show that DNA repair genes of the type of mismatch repair genes and few others are frequently mutated in colorectal tumors with high number of point mutations and low level of aneuploidy. Instead another class of mutation drivers, that they define apoptotic genes are more frequently mutated in highly aneuploid tumors and in tumor types different than gastro intestinal and uterine. The authors also claimed the mutation signatures found in this work possess substantial prognostic power.

Although the analysis of the different types of driver genes mutated in high and low aneuploidy tumors is interesting, it needs further exploration and validation (see especially the IMPORTANT POINT highlighted below). In addition, more than half of the paper focuses on the fact that gastrointestinal and uterine tumors can be classified in 2 subtypes, high point mutations/low aneuploidy and low point mutations/high aneuploidy and the fact that this has important prognostic implications. Although interesting (especially because we do not really know the mechanism of this mutual exclusivity), this observation is not new.

Major points:

1. The fact that aneuploidy anti correlates with point mutations in colorectal, uterine and few other tumors but positively correlates with aneuploidy (Fig. 1) in most of the other cancers has already been shown by the Elledge group (Davoli et al 2017, Fig. 1).

2. Fig.1: In Fig. 1b (top panels) and several other plots (for example Fig. S1) the authors pool together different tumor types to draw survival plots. They pool together all the other tumor types for the survival plot in Fig. 1b (**top panels**). The authors should plot survival curves independently in each tumor type. Although they show the result of the KP analysis in 1a lower panel for individual tumor types (and many tumors are not significant), pooling together all the tumor samples in 1b does not make sense and is statistically not correct (obviously different tumor types have different average survival time). The authors should add hazard ratios to the survival plots. Also, in 1b top panels, the split between tumors with driver mutations $>$ or $<$ 1 has a significant association with survival and decent HR but the HR seems to strongly decrease (as well as the significance) with threshold of 2 or 3 or more mutations.

The survival association shown in 1b lower plot is stronger, meaning tumors with high number of mutations (or driver mutations) are associated with better prognosis than tumors with lower N of mutations. This has been known for a long time in these tumors (see for example: <https://www.ncbi.nlm.nih.gov/pmc/articles/PMC5938532/>) and is not novel.

3. Fig. 2: In Fig. 2a, the authors should split the gastrointestinal and uterine tumors in the MSI (that tend to correspond to high mutations) and MSS (~low mutations). They

should repeat the analysis for the 2 classes of tumors. With this distinction the authors will probably find that the MSS tumors of gastrointestinal, uterine tumors have mutations in TP53 and the other MSS class of genes, more similar to the rest of the tumor types. Without this distinction Fig. 2a is over-simplified and it seems that all the gastrointestinal/uterine and similar tumors are associated with mismatch repair gene mutations and other similar mutations.

The authors improperly use the word signature. Mutation signature has a specific meaning (see papers by Alexandrov et al.). In this paper the authors use this term with another meaning and so they should use another word such 'enrichment' or 'gene set enrichment' or similar.

The authors use the word 'apoptosis' referring to genes that are not necessarily related to apoptosis but to DNA damage repair and checkpoint (see sentence: ' and the associated apoptosis factors, such as BCL3, BRCA2, CHEK2, PML, TOPORS, TP63, AEN and SIRT1'). Examples of these genes related to DNA damage checkpoint more than apoptosis are TP53, BRCA2 (the 2 most strongly associated with aneuploidy), CHECK2, TOPORS, BARD1. So although the result of their pathway analysis are gene sets called 'P53 apoptotic signaling damage response' (Fig. 2c), since we know the function of most of these genes, it is not correct to call them apoptotic genes, but the authors should call them 'genes involved in DNA damage checkpoint signaling'. The title should be change accordingly. For example one of the most frequently mutated true apoptotic gene in cancer is Casp8: this is a real apoptotic gene, not a DNA damage gene. This gene does not come out in the analysis further supporting the fact that the enrichment the authors describe is for genes implicated not (or not only) in apoptosis but in DDR checkpoint.

For Fig. 2a, the authors should show the FDR, not the p-value.

4. Fig. 3

In Fig. 3c, the gene sets that anticorrelate with aneuploidy is quite clear and is represented by genes acting in fixing point mutations such as mismatch repair genes. On the other hand the genes whose mutation is positively correlated with aneuploidy have a much weaker signal (3c, right panel and Fig. S1a). The only gene that has a really strong effect is TP53. BRCA2 and TP63 also have a relatively strong signature. However, the other genes in the signature, albeit significant do not have a strong effect. Although their p-value is significant, their association appears to be substantially weaker than the top 3 genes (Fig. 3c and Fig. S1a).

IMPORTANT POINT:

Given that TP53 is the strongest gene showing positive association with aneuploidy (when mutated), are the result shown in Fig. 3a and b dependent on TP53 mutation? What happens if the authors calculate the ratio excluding TP53? And perhaps also BRCA2? If it is really the entire gene sets that contribute to this signature, the signal should persist even after ignoring TP53 (and BRCA2).

5. Fig. 4/5

Fig. 4 and 5 reflect the fact that for gastrointestinal and uterine tumors high mutations anti correlate with aneuploidy and that tumors with high mutations and low aneuploidy are associated to better prognosis than the other tumors. As mentioned above multiple times, this is known.

Other points:

1. In this work, the aneuploidy score is arm-level. How about chromosome- and focal-level aneuploidy? Do they have same or different relationship with driver mutations?

2. In Figure 2, the author found two distinctive mutational signatures in DNA damage response pathways which can distinguish gastrointestinal and endometrial cancers from all other cancers. To do this the authors start from genes involved in DNA damage. What would be the result if the authors used all genes? Would they find other pathways?

3. In Figure 1d, most driver genes of PAAD show negative correlation with aneuploidy. Why the driver mutations load is highly positive correlated with aneuploidy as shown in Figure 1a?

4. In figure 2b, the repair and apoptosis signature show positive correlation with driver mutations load in gastrointestinal and endometrial tumors and other tumors respectively. If arbitrary gene sets of the same size are selected, are they also positive correlated with driver mutations load or not?

5. In figure 3a, how exactly is the ratio of the apoptosis to repair signatures load calculated?

6. Which type of tumor is shown in figure 3c or is it a plot representing all tumor types?

Reviewers' comments:

Reviewer #2 (Remarks to the Author): Expertise in aneuploidy and bioinformatics

Auslander et al. explore the relationship between driver mutations and chromosomal alterations. They identify certain tumours (generally hyper-mutator) which exhibit many mutations, but little aneuploidy.

This manuscript has major issues which preclude recommendation for publication as it stands.

Comments:

- The authors appear to define driver mutations as any mutation in a cancer gene. Many driver mutations will therefore be false positives.

-Relatedly, this means that their measure of number of driver mutations may in fact capture mutation burden rather than driver burden. Thus, relationships with outcome may reflect number of neoantigens rather than number of driver mutations. For instance, the authors state that: "although in most tumor types, high number of driver mutations is expectedly associated with poor outcome, gastrointestinal and endometrial tumors show an inverse relationship (Fig. 1a-c)." This likely reflects the good prognosis associated with hypermutator tumours (which is known).

- The authors should repeat the analysis controlling for mutation burden. Are the results still significant?

-I found the DDR mutational signatures predicting driver mutation load rather confusing. A figure describing what is being done would be useful.

-The p-value cut-offs seem to vary and often were not very stringent - e.g. $P < 0.1$.

-The identification of MSI tumours with better prognosis does not seem particularly novel?

- The number of samples used for different TCGA subsets seems strange. This is not all TCGA samples?

- The code needs to be made available now - not in the future.

-The authors should consider repeating the analysis, but separating gains and losses

-How different would the results be if the authors used mutational signatures to define MSI?

REVIEWER #1

In this manuscript, Auslander and colleagues analyzed the relationship between aneuploidy and point mutations across tumors. They find that in colorectal, uterine, stomach and few other tumors, aneuploidy is strongly anti-correlated with mutation load. On the contrary, for the other tumor types, aneuploidy and point mutations are positively associated. They go on to show that DNA repair genes of the type of mismatch repair genes and few others are frequently mutated in colorectal tumors with high number of point mutations and low level of aneuploidy. Instead another class of mutation drivers, that they define apoptotic genes are more frequently mutated in highly aneuploid tumors and in tumor types different than gastro intestinal and uterine. The authors also claimed the mutation signatures found in this work possess substantial prognostic power. Although the analysis of the different types of driver genes mutated in high and low aneuploidy tumors is interesting, it needs further exploration and validation (see especially the IMPORTANT POINT highlighted below). In addition, more than half of the paper focuses on the fact that gastrointestinal and uterine tumors can be classified in 2 subtypes, high point mutations/low aneuploidy and low point mutations/high aneuploidy and the fact that this has important prognostic implications. Although interesting (especially because we do not really know the mechanism of this mutual exclusivity), this observation is not new.

Major points:

1. The fact that aneuploidy anti correlates with point mutations in colorectal, uterine and few other tumors but positively correlates with aneuploidy (Fig. 1) in most of the other cancers has already been shown by the Elledge group (Davoli et al 2017, Fig. 1).

Several studies have indeed reported similar results for point mutations and aneuploidy (either chromosomal or focal SCNA), including Davoli et al 2017 (which is now cited), as well as Buccitelli et al 2017 and Taylor et al 2018 (cited in the previous and revised version). However, as shown in Figure 1, here, we aim to investigate the associations between arm-level aneuploidy and the mutation load in known cancer driver genes (not the overall mutation load), and explore the associations of both types of genomic changes with survival. To the best of our knowledge, this was not the focus of any of the previous studies. This is now explained in the manuscript:

“First, we analyzed the correlation between the number of mutations¹⁴ in cancer driver genes (which is used as a proxy for the number of actual driver mutations) and aneuploidy levels in each tumor type. In agreement with the previous observations for the overall mutational load^{10,11}, the correlations were positive for most tumor types, but

significantly negative for gastrointestinal and endometrial tumors in which we also noticed a higher load of driver mutations (Fig. 1a). We next investigated the association between the number of driver mutations and overall survival rates. We found that, although in most tumor types, a large number of driver mutations is predictably associated with poor outcome, most of the gastrointestinal and endometrial tumors show an inverse relationship (Fig. 1a-b). This trend is recapitulated with aggregated data from these two classes for tumor types although the different survival rates in different tumor types are likely to be a confounding factor in this analysis (Fig. 1c). However, examination of survival association with the overall mutation load reveals positive association, mostly, in hypermutated tumors (including those with a negative association between driver mutations and survival, such as lung carcinomas; Supplementary Fig. 1)..” [page 4]

2. Fig.1: In Fig. 1b (top panels) and several other plots (for example Fig. S1) the authors pool together different tumor types to draw survival plots. They pool together all the other tumor types for the survival plot in Fig. 1b (top panels). The authors should plot survival curves independently in each tumor type. Although they show the result of the KP analysis in 1a lower panel for individual tumor types (and many tumors are not significant), pooling together all the tumor samples in 1b does not make sense and is statistically not correct (obviously different tumor types have different average survival time). The authors should add hazard ratios to the survival plots.

We agree with this comment. We now show the hazard ratio instead of KM delta AUC, and replaced Figure 1b (previously, 1c) to show the KM hazard ratio and P-value for individual tumor types, for 5 different thresholds of mutations in driver genes. However, we still find it of interest to show the aggregate of the two clusters for two thresholds. These aggregated data are shown in Figure 1c, but we explicitly mention in the main text the limitations of the analysis that involves pooling together different tumor types:

“This trend is recapitulated with aggregated data from these two classes for tumor types although the different survival rates in different tumor types are likely to be a confounding factor in this analysis (Fig. 1c).” [page 4]

Also, in 1b top panels, the split between tumors with driver mutations $>$ or $<$ 1 has a significant association with survival and decent HR but the HR seems to strongly decrease (as well as the significance) with threshold of 2 or 3 or more mutations.

This is likely due to the different numbers of driver mutations in these cohorts (where the median number of driver mutations in the top panel is lower, Fig 1A). We now show the hazard ratio for each tumor, for several numbers of driver mutations.

The survival association shown in 1b lower plot is stronger, meaning tumors with high number of mutations (or driver mutations) are associated with better prognosis than tumors with lower N of mutations. This has been known for a long time in these tumors (see for example: <https://www.ncbi.nlm.nih.gov/pmc/articles/PMC5938532/>) and is not novel.

When evaluating mutation load in cancer driver genes, this trend holds mainly for gastrointestinal and endometrial tumors. For most of the other tumors (even some hyper-mutated ones, such as lung tumors), high load of driver mutations is associated with poor survival (In new Fig. 1b). In contrast, when considering all mutations, there is indeed a positive association between overall mutation load and survival in most hyper-mutated tumors (as shown previously, Supp. Fig 1). However, the associations between survival and the overall mutation load in gastrointestinal and endometrial tumors is not as strong as that of driver mutation load and survival (Supp. Fig. 1). This is now explained in the main text:

“However, examination of survival association with the overall mutation load reveals positive association, mostly, in hypermutated tumors (including those with a negative associations between driver mutations and survival, such as lung carcinomas, Supplementary Fig. 1), consistent with previous findings¹⁶. Furthermore, these associations are reproduced when controlling for the total mutation burden, and when considering whole chromosome aneuploidy or separately evaluating for arm gains and losses; together, these observations further support the unique associations characteristic of gastrointestinal and endometrial tumors (Supplementary Fig. 2 and 3).”
[page 4]

3. Fig. 2: In Fig. 2a, the authors should split the gastrointestinal and uterine tumors in the MSI (that tend to correspond to high mutations) and MSS (~low mutations). They should repeat the analysis for the 2 classes of tumors. With this distinction the authors will probably find that the MSS tumors of gastrointestinal, uterine tumors have mutations in TP53 and the other class of genes, more similar to the rest of the tumor types. Without this distinction Fig. 2a is over-simplified and it seems that all the gastrointestinal/uterine and similar tumors are associated with mismatch repair gene mutations and other similar mutations.

We appreciate this comment. In the analysis in Figure 2a, we aim to specifically identify mutations that are associated with high driver mutation load in gastrointestinal and endometrial tumors. Indeed, this figure does not indicate that these tumors are associated with MMR genes in general, but only demonstrate this association for high driver mutations load in these tumors (this is expected because high load of driver mutations is associated with MSI in these tumors).

In response to this comment, we now show the mutations in the two selected gene sets (repair and apoptosis) vs. the load of driver mutations, separately for MSI and MSS gastrointestinal and endometrial tumors, and also for all gastrointestinal and endometrial tumors (Supp. Fig 10). As can be seen in this figure, it is actually the MSS tumors that show the stronger negative correlation between TP53 and driver mutational load, and stronger positive correlation between MMR genes and driver mutational load. This is likely the case because MSI tumors tend to have high mutation loads in many different genes as well as many random mutations, hence it is difficult to recognize such patterns in those tumors.

The authors improperly use the word signature. Mutation signature has a specific meaning (see papers by Alexandrov et al.). In this paper the authors use this term with another meaning and so they should use another word such 'enrichment' or 'gene set enrichment' or similar.

We agree with this comment. To avoid the confusion around the term "mutation signature", in the revision, we use wording such as "distinctive sets of mutated genes" instead.

The authors use the word 'apoptosis' referring to genes that are not necessarily related to apoptosis but to DNA damage repair and checkpoint (see sentence: '...and the associated apoptosis factors, such as BCL3, BRCA2, CHEK2, PML, TOPORS, TP63, AEN and SIRT1'). Examples of these genes related to DNA damage checkpoint more than apoptosis are TP53, BRCA2 (the 2 most strongly associated with aneuploidy), CHECK2, TOPORS, BARD1. So although the result of their pathway analysis are gene sets called 'P53 apoptotic signaling damage response' (Fig. 2c), since we know the function of most of these genes, it is not correct to call them apoptotic genes, but the authors should call them 'genes involved in DNA damage checkpoint signaling'. The title should be change accordingly. For example one of the most frequently mutated true apoptotic gene in cancer is Casp8: this is a real apoptotic gene, not a DNA damage gene. This gene does not come out in the analysis further supporting the fact that the enrichment the authors describe is for genes implicated not (or not only) in apoptosis but in DDR checkpoint.

We appreciate this comment. The reason for calling this gene set 'apoptosis' was that the pathways specifically enriched with mutated genes in this set are almost all apoptosis-related (Fig 2c). Indeed, these genes are mostly components of DDR-related checkpoints, which is mentioned in several places in the revised manuscript. To further validate that this gene set is indeed enriched in apoptotic pathways, we now additionally show the STRING pathway enrichment analysis of these sets, for GO, KEGG and REACTOM pathway enrichment analyses (Supp. Fig 5). As can be seen from this new figure, the top ranked pathways are consistent, and are DNA repair pathways for the repair gene set, and apoptotic pathways for the other gene set. Hence, although P53 and BRCA2 as well as several other genes are indeed linked to DNA damage checkpoints rather to apoptosis as such, the overall context of this gene set appears to be more strongly associated with apoptosis and, specifically, with P53-dependent apoptotic signaling pathways. We now acknowledge both possibilities across the manuscript, but maintain the name of this gene set because of these results and for the sake of brevity.

For Fig. 2a, the authors should show the FDR, not the p-value.

For a given tumor type, the displayed P-value is binomial, evaluating the rank of the score assigned to each gene. Thus, for a given tumor type, there is no need for FDR or any correction (as each DDR gene is assigned with a P-value by its rank, one hypothesis is evaluated, with the top 5% of the genes considered significant). When combining the scores across multiple tumor types in a cluster, for each gene, the assigned P-values are combined into a single test statistic (X^2) using Fisher's method (which is standard adjustment for combining multiple P-values). We provide these Fisher P-values in Supp. Table 4, and now we also show them in Fig2a. We nevertheless believe that the individual P-values assigned to each tumor type before adjustment are of interest as well.

4. Fig. 3

In Fig. 3c, the gene sets that anticorrelate with aneuploidy is quite clear and is represented by genes acting in fixing point mutations such as mismatch repair genes. On the other hand the genes whose mutation is positively correlated with aneuploidy have a much weaker signal (3c, right panel and Fig. S1a). The only gene that has a really strong effect is TP53. BRCA2 and TP63 also have a relatively strong signature. However, the other genes in the signature, albeit significant do not have a strong effect. Although their p-value is significant, their association appears to be substantially weaker than the top 3 genes (Fig. 3c and Fig. S1a).

IMPORTANT POINT: Given that TP53 is the strongest gene showing positive association with aneuploidy (when mutated), are the result shown in Fig. 3a and b dependent on TP53 mutation? What happens if the authors calculate the ratio excluding TP53? And perhaps also BRCA2? If it is really the entire gene sets that contribute to this signature, the signal should persist even after ignoring TP53 (and BRCA2).

We appreciate this comment. We now repeat the analysis underlying Figure 3, but after excluding TP53 and BRCA2 from the apoptosis mutated gene set (shown in Supp. Fig. 9). As can be observed from this figure, the overall conclusion does not change, and the ratio between the apoptosis and repair mutated gene sets remains positively correlated with aneuploidy across tumor types. However, when these genes are excluded, the apoptosis set load is negatively correlated with aneuploidy in gastrointestinal and endometrial tumors (the correlation of this set with aneuploidy remains positive for most of the other tumor types).

As shown in both the revised and the original versions (Supp Figs 1 and 8) of our manuscript, the only gene that is significantly and positively correlated with aneuploidy (and negatively correlated with MSI) in gastrointestinal and endometrial tumors is TP53 (APC is positively correlated but the correlation is not significant).

We added panels to Supp Fig. 8 (now a and b) showing that, similarly, the only gene that is negatively correlated with the driver mutational load in these tumors is TP53, whereas for other tumor types, it is one of the genes with the strongest positive correlation with the driver mutational load. Hence, it is naturally selected to predict driver load in the second cluster of tumors, and is a substantial component of the selected mutated gene set for that cluster (the apoptosis gene set).

In summary, indeed, TP53 is the component that sustains the negative correlation between aneuploidy and the apoptosis signature, but this is true only for the gastrointestinal and endometrial tumors. In addition, it is crucial for the positive correlation with the driver mutation load in the cluster where it was selected. Although any gene set excluding TP53 shows a negative correlation with aneuploidy across gastrointestinal and endometrial tumors, the ratio between the mutations in the two sets remains positively correlated with aneuploidy. These observations are discussed in the revised manuscript:

“Indeed, TP53 shows the strongest positive association with aneuploidy as the only gene that is positively and significantly associated with aneuploidy in gastrointestinal and endometrial tumors (Supplementary Fig. 8). Nevertheless, excluding TP53 (as well as BRCA2) from the apoptosis mutated gene set does not eliminate the association of the ratio between the repair and apoptosis sets with aneuploidy (Supplementary Fig. 9).” [page 9]

5. Fig. 4/5

Fig. 4 and 5 reflect the fact that for gastrointestinal and uterine tumors high mutations anti correlate with aneuploidy and that tumors with high mutations and low aneuploidy are associated to better prognosis than the other tumors. As mentioned above multiple times, this is known.

We appreciate this comment and regret that this has not been properly clarified in the original version of the manuscript. We are aware of the fact that high mutation rate, MSI and low aneuploidy have been previously associated with improved survival for gastro-intestinal and endometrial tumors, and that the anti-correlation between MSI and aneuploidy has been reported (mostly, for colorectal tumors). We cited several references reporting such findings, and more have been added in the revised version. However, we believe that our analysis includes substantial novelty which, unfortunately, we failed to properly emphasize in the original manuscript. Specifically:

1. In figure 4, we show that a newly derived set of mutations captures the mutual exclusivity of aneuploidy and MSI across all gastrointestinal and endometrial tumors. We show that this set robustly predicts MSI in several independent datasets and is strongly associated with survival in gastrointestinal and endometrial tumors (more strongly than either aneuploidy or MSI are alone). In addition, we show the mutual exclusivity pattern between chromosomal aneuploidy and MSI for TCGA tumors which, to the best of our knowledge, has not been demonstrated for such a large cohort previously (rather, this exclusivity was implied specifically for colorectal tumors).

2. In Figure 5a, we show that the association of MSI and survival is likely explained by the association of arm-level aneuploidy and survival, and the mutual exclusivity with MSI. We show that there is little difference in survival between MSS or MSI diploid tumors. Hence, it is possible that MSI is associated with improved survival simply because of the mutual exclusivity with aneuploidy (as aneuploidy is associated with poor survival, which is known from previous studies). This as well, to the best of our knowledge, has not been investigated before with such large sample set. Figure 5b-c shows that the mutations in the MSI-aneuploidy set, that captures this mutual exclusivity, are predictive of survival in gastrointestinal and endometrial tumors to a greater extent than either MSI or aneuploidy alone. Figure 5d-f explores associations of aneuploidy, MSI and the MSI-aneuploidy mutational set with drug response for the TCGA cohort all of which, to the best of our knowledge, has not been shown previously.

We edited this part of the manuscript in the revised version, to first mention all that has already been reported, and then explicitly explain the added value in the

present analyses, and provide the motivation for these.

Other points:

1. In this work, the aneuploidy score is arm-level. How about chromosome- and focal-level aneuploidy? Do they have same or different relationship with driver mutations?

We appreciate this comment. In the revised manuscript, we show the relationship between driver mutations load and (a) whole chromosomal aneuploidies, (b) arm-level gains and (c) arm level losses (Supp. Fig 3). As can be observed, the two relationships are highly similar.

While this is beyond the scope of the present work, which focuses on non-focal arm aneuploidies, we used the data from Davoli et al 2017 (which provides focal SCNA levels for 13 of the tumor types studied). We observe that the associations of driver mutation load and focal SCNA differ from those with arm levels aneuploidies (For example, SKCM shows significant negative correlation, Supp. Fig 3b). Although this analysis is limited to a subset of the tumor types, precluding a confident conclusion, it appears that focal-level SCNA could have a different relationship with driver mutations.

2. In Figure 2, the author found two distinctive mutational signatures in DNA damage response pathways which can distinguish gastrointestinal and endometrial cancers from all other cancers. To do this the authors start from genes involved in DNA damage. What would be the result if the authors used all genes? Would they find other pathways?

In the revision, we repeat this analysis starting with all genes (Supp. Fig. 13). In this analysis, the set selected for gastrointestinal and endometrial cancers is not enriched with DNA repair genes (but mostly developmental and growth related genes), likely, because the tumors with impaired mismatch repair contain mutations in many different genes, thus making in difficult to identify the initial set of mutated DNA repair genes. By contrast, the set selected for other tumor types is still enriched with apoptotic signaling and senescence. Importantly, the ratio between these sets is also associated with survival and aneuploidy across tumor types, similarly to the originally selected sets. We write in the revised manuscript:

“Repeating this analysis without limiting the search for enriched mutated gene sets to DDR genes (i.e. starting from all genes) did not yield a DNA-repair enriched set of genes for the gastrointestinal and endometrial tumor. This is likely to be the case because tumors with impaired mismatch repair contain mutations in many different genes, thus making in difficult to identify the initial set of mutated DNA repair genes. By contrast, the set selected for other tumor

types was still enriched with genes involved in apoptotic pathways. Crucially, the ratio between these sets showed similar associations with aneuploidy and overall survival rates as the DDR-limited search (Supplementary Fig. 13).”

3. In Figure 1d, most driver genes of PAAD show negative correlation with aneuploidy. Why the driver mutations load is highly positive correlated with aneuploidy as shown in Figure 1a?

This is because the PAAD driver mutation load is generally very low, and accordingly, so are the frequencies of most of the driver mutations in PAAD. The only driver mutations with high frequency in PAAD are KRAS and TP53 (observed in 70.8% and 60.8% of the samples, respectively). No other driver gene has mutation frequency over 25% in PAAD (SMAD4 and CDKN2A have 21.7% and 19.2%, respectively, and all other genes have less than 7%, see the plot below). Excluding KRAS and TP53, 57% of the PAAD sample contain one or zero mutations in cancer driver genes. Hence, the load of mutation in cancer driver genes in PAAD is mostly affected by KRAS and TP53, which are both strongly, positively correlated with aneuploidy.

4. In figure 2b, the repair and apoptosis signature show positive correlation with driver mutations load in gastrointestinal and endometrial tumors and other tumors respectively. If arbitrary gene sets of the same size are selected, are they also positive correlated with driver mutations load or not?

The correlations between driver mutations load and randomly selected gene sets from either all genes or DDR genes are shown in Supp. Fig. 6. Mostly, arbitrary

mutation sets show substantially lower correlation with the load of driver mutations as pointed out in the revised manuscript. However, a fraction of the randomly selected DDR gene sets shows correlation with driver mutations in gastrointestinal and endometrial tumors, likely, because of the overall correlation between driver mutations and DDR genes in MMR deficient tumors.

5. In figure 3a, how exactly is the ratio of the apoptosis to repair signatures load calculated?

It is the number of mutations in the apoptosis mutated gene set divided by the number of mutations in the repair gene set.

6. Which type of tumor is shown in figure 3c or is it a plot representing all tumor types?

Figure 3c shows the top and bottom 0.05 quartiles of the apoptosis to repair set load ratios, when considering all tumor types.

Reviewer #2 (Remarks to the Author): Expertise in aneuploidy and bioinformatics

Auslander et al. explore the relationship between driver mutations and chromosomal alterations. They identify certain tumours (generally hyper-mutator) which exhibit many mutations, but little aneuploidy.

This manuscript has major issues which preclude recommendation for publication as it stands.

Comments:

- The authors appear to define driver mutations as any mutation in a cancer gene. Many driver mutations will therefore be false positives.

It is true that we define driver mutation as non-silent mutations in established cancer driver gene. We are unaware of any approach to identify the mutations that truly drove the development of a given tumor, using "snapshot" genomic data. This limitation is now mentioned in the main text:

"First, we analyzed the correlation between the number of mutations¹⁴ in cancer driver genes (which is used as a proxy for the number of actual driver mutations) and aneuploidy levels in each tumor type." [page 4]

-Relatedly, this means that their measure of number of driver mutations may in fact capture mutation burden rather than driver burden. Thus, relationships with outcome may reflect number of neoantigens rather than number of driver mutations. For instance, the authors state that:

"although in most tumor types, high number of driver mutations is expectedly associated with poor outcome, gastrointestinal and endometrial tumors show an inverse relationship (Fig. 1a-c)." This likely reflects the good prognosis associated with hypermutator tumours (which is known).

We appreciate this comment. In the revision, we repeat the analysis of Figure 1a-c, but with the overall mutation load (Supp. Fig. 1). As can be seen from this figure, indeed, the most hyper-mutated tumors (LUAD, LUSC and SKCM) show better prognosis for the higher overall mutation load. However, these tumors do not show negative associations between the mutation load and chromosomal-arm aneuploidy, and lung tumors (LUAD and LUSC) show better prognosis for lower driver mutation load. It is specifically the tumor types with the highest load of mutations in driver genes (gastrointestinal and endometrial, which are not those with the highest overall mutation load) that show negative associations of the driver genes mutation load with both chromosomal aneuploidy and survival.

- The authors should repeat the analysis controlling for mutation burden. Are the results still significant?

We appreciate this helpful suggestion. We repeated the analysis, controlling for the total mutation burden (separating samples by range of mutation burden, Supp. Fig. 2). As can be seen from this figure, gastrointestinal and endometrial tumors maintain a similar pattern for low, medium and high overall mutational burden. We observed some variation for a few of the other tumor types, but the overall conclusion stands.

-I found the DDR mutational signatures predicting driver mutation load rather confusing. A figure describing what is being done would be useful.

A figure describing the process employed to derive these signatures (now called distinctive sets of mutated genes as per the comments of reviewer 1) has been added as Supp. Fig 11

-The p-value cut-offs seem to vary and often were not very stringent - e.g. $P < 0.1$.

We changed all P-values cutoffs to 0.05, except for the cutoff for the selection of gene sets, as this is used only as a threshold and not for significance evaluation (and we additionally report the results for using $P < 0.05$ for selecting these sets in the supplementary material).

-The identification of MSI tumours with better prognosis does not seem particularly novel?

We appreciate this comment and regret that this has not been made sufficiently clear in the original manuscript. Indeed, it has been reported previously that MSI tumors are associated with better prognosis. The novelty of the analysis performed for figures 4-5 is explained here and as a response to referee 1, and is now clarified in the revised manuscript:

1. In figure 4 we show that a newly derived set of mutation captures the mutual exclusivity of aneuploidy and MSI across all gastrointestinal and endometrial tumors. We show that this set robustly predicts MSI in several independent datasets, and is highly associated with survival in gastrointestinal and endometrial tumors (more than either aneuploidy or MSI are alone). In addition, we show the mutual exclusivity pattern between chromosomal aneuploidy and MSI for TCGA tumors, which, to the best of our knowledge, was not shown for such a large data cohort previously (and was mainly implicated for colorectal tumors).

2. In Figure 5a we show that the association of MSI and arm-level aneuploidy with survival is likely just a result of the association with aneuploidy and not MSI. This is because there is no noticeable difference in survival between MSS or MSI diploid tumors. Hence, it is possible that MSI is associated with improved survival simply because of the mutual exclusivity with aneuploidy (as aneuploidy is associated with poor survival). This, to the best of our knowledge, has not been investigated before. Figure 5b-c shows that the mutations in the MSI-aneuploidy set that captures this mutual exclusivity are better predictors of survival in gastrointestinal and endometrial tumors than MSI or aneuploidy alone. Figure 5d-f explores associations of aneuploidy, MSI and the MSI-aneuploidy mutational set with drug response for the TCGA cohort, all of which, to the best of our knowledge, has not been shown previously.

- The number of samples used for different TCGA subsets seems strange. This is not all TCGA samples?

For analysis in Figures 1-3, all TCGA samples of primary, solid tumors with available mutational, CNA (aneuploidy) and clinical data were considered (8686 samples overall). No further filtering was applied. For analysis of Figure 4-5, all

Gastrointestinal and endometrial tumors with (1) available mutational, CNA (aneuploidy) and clinical data and (2) MSI information were considered.

- The code needs to be made available now - not in the future.

The code is provided as a private GitHub repository in:

noamaus/INTERPLAY-TUMOR-CODES

It can be freely accessed by the referees using the login information:

username: INTERPLAY-referees

password: INTERPLAYCODES123

This repository will be made publicly available upon publication (and will be updated to include the additional code written to perform the most recent analyses in response to the referees comments).

-The authors should consider repeating the analysis, but separating gains and losses

We now show the relationship between driver mutations load and (a) arm-level gains and (b) arm level losses (Supp. Fig 3) and (c) whole chromosomal aneuploidies. As can be observed, the relationships are almost identical to that observed for arm-level aneuploidies.

-How different would the results be if the authours used mutational signatures to define MSI?

Unfortunately, we do not understand this comment.

Reviewers' comments:

Reviewer #1 (Remarks to the Author):

The authors have edited the manuscript according to our comments.

A few points:

- In fig1 the HR is shown ranging from -0.5 and 0.5. Usually HR is centered around 1. Is this because the authors log transform the HR? If so, they should specify. (I could have missed this and apologize if this is the case.)
- A weak point of the paper that remains is the at least partial lack of novelty in the aneuploidy level and point mutations also with respect to survival; the authors now acknowledge this.
- In Fig. S8 the authors show now the same analysis shown in Fig. 3 after removing BRCA2 and TP53. The result does change at least in part, the authors acknowledge this. Can the In Fig. 3 I do not see the asterisk for significance that is mentioned in the legend, they should add that.

Reviewer #2 (Remarks to the Author):

The authors have improved the manuscript, but it still requires work, and the novelty still appears to be lacking somewhat. In particular, the authors must demonstrate their results go beyond findings related to mutational burden and SCNA.

- there are a lot of methods for attempting to define driver mutations as true or false positives. The authors could consider evaluating predictors of functional impact, such as PolyPhen or SIFT scores, or more specific tools for cancer genes - such as those outlined in the Bailey Cell paper. The authors could also consider running dNdScv to predict the proportion of alterations in their cancer genes that are likely selected and therefore not passengers.
- The authors should also consider if they observe similar relationship when they focus on SCNA driver alterations.
- separating samples by range of mutation burden is not the same as controlling for mutation burden as a continuous variable. The authors should properly account for mutation burden.
- there are many approaches and tools for defining mutational signatures - including those described by Alexandrov. The authors should explore whether their results remain consistent if they define MSI groups according to mutational signatures.

Reviewers' comments:

Reviewer #1 (Remarks to the Author):

The authors have edited the manuscript according to our comments.

A few points:

- In fig1 the HR is shown ranging from -0.5 and 0.5. Usually HR is centered around 1. Is this because the authors log transform the HR? If so, they should specify. (I could have missed this and apologize if this is the case.)

We apologize for the confusion. Indeed, these scores are log10 transformed (to conform with the range of the correlation coefficient). This is now indicated in the figure legends.

- A weak point of the paper that remains is the at least partial lack of novelty in the aneuploidy level and point mutations also with respect to survival; the authors now acknowledge this.

- In Fig. S8 the authors show now the same analysis shown in Fig. 3 after removing BRCA2 and TP53. The result does change at least in part, the authors acknowledge this. Can the In Fig. 3 I do not see the asterisk for significance that is mentioned in the legend, they should add that.

Unfortunately, this comment seems to be incomplete. Figure 3 does show the asterisks.

Reviewer #2 (Remarks to the Author):

The authors have improved the manuscript, but it still requires work, and the novelty still appears to be lacking somewhat. In particular, the authors must demonstrate their results go beyond findings related to mutational burden and SCNA.

- there are a lot of methods for attempting to define driver mutations as true or false positives. The authors could consider evaluating predictors of functional impact, such as polyphen or SIFT scores, or more specific tools for cancer genes - such as those outlined in the Bailey Cell paper. The authors could also consider running dNdScv to predict the proportion of alterations in their cancer genes that are likely selected and therefore not passengers.

In the revised manuscript, we included results obtained with (1) polyphen and (2) SIFT scores, to filter out possible false positive. We repeated the analysis shown in Figure 1, where for each sample we count the number of mutations in cancer driver genes that are (1) scored as 'damaging' (non-benign) via PolyPhen and (2) scored as deleterious (not tolerated) via SIFT scores (Supp. Fig. 2). The findings remain similar when using both these scores to count the number of cancer driver mutations. However, we believe that the dNdScv analysis is outside the scope of this work.

- The authors should also consider if they observe similar relationship when they focus on SCNA driver alterations.

Because most SCNA alterations are highly and positively correlated, SCNA alterations in drivers and aneuploidy are highly, positively correlated across all tumors, and are, mostly, associated with poor survival (see figure below). Hence, the associations observed in this work are unique for point mutations. We do not believe that including this analysis in the manuscript would substantially add to it.

- separating samples by range of mutation burden is not the same as controlling for mutation burden as a continuous variable. The authors should properly account for mutation burden.

We agree with this comment. We now replaced the original analysis with the suggested analysis controlling for the mutational burden as a continuous variable, using partial correlations and cox-regression analysis (Supp. Fig. 3). The conclusions remain similar, but interestingly, the partial correlations between the number of driver mutations and aneuploidy (controlling for overall mutational load) is negative for several additional tumor types such as head and neck, and thyroid carcinomas, are significantly positive only for tumor types with very low overall number of mutations in cancer driver genes.

- there are many approaches and tools for defining mutational signatures - including those described by Alexandrov. The authors should explore whether their results remain consistent if they define MSI groups according to mutational signatures.

We appreciate this comment. We repeat the analyses reported in Figure 4 and 5a with MSI defined by the mutational signatures described by Alexandrov (Cosmic

signatures 6,14,15,20,21 and 26, which are associated with defective DNA mismatch repair and MSI, Supp. Fig 11). We find that, although, as expected, the correlation between aneuploidy and these signatures is negative, it is not always significant; the most highly significant result was obtained for the COSMIC signature 6. Importantly, we find that these signatures are associated with poor survival in diploid tumors (Supp. Fig. XX), supporting our conclusion that the positive associations between MSI and survival could be merely due to the negative associations between MSI and aneuploidy.

REVIEWERS' COMMENTS:

Reviewer #1 (Remarks to the Author):

In the revised version the authors have successfully addressed my concerns and comments.

Reviewer #2 (Remarks to the Author):

The authors have improved the manuscript. However, novelty is still somewhat lacking. As mentioned in the previous review, the manuscript could be further improved if the authors could apply dNdScv to calculate number of expected driver alterations within each cancer type (the authors could use the results from Martincorena et al., 2017).